# Computer State Evaluation Using Adaptive Neuro-Fuzzy Inference Systems

**DOI:** 10.3390/s22239502

**Published:** 2022-12-05

**Authors:** Abror Buriboev, Azamjon Muminov

**Affiliations:** 1Department of IT, Samarkand Branch of Tashkent University of Information Technologies, Samarkand 140100, Uzbekistan; 2Department of Computer Engineering, Gachon University, 7 Sujeong-gu, Seongnam-si 461-701, Republic of Korea

**Keywords:** Mamdani and Sugeno adaptive neuro-fuzzy inference system, CPU utilization, complex evaluation

## Abstract

Several crucial system design and deployment decisions, including workload management, sizing, capacity planning, and dynamic rule generation in dynamic systems such as computers, depend on predictive analysis of resource consumption. An analysis of the computer components’ utilizations and their workloads is the best way to assess the performance of the computer’s state. Especially, analyzing the particular or whole influence of components on another component gives more reliable information about the state of computer systems. There are many evaluation techniques proposed by researchers. The bulk of them have complicated metrics and parameters such as utilization, time, throughput, latency, delay, speed, frequency, and the percentage which are difficult to understand and use in the assessing process. According to these, we proposed a simplified evaluation method using components’ utilization in percentage scale and its linguistic values. The use of the adaptive neuro-fuzzy inference system (ANFIS) model and fuzzy set theory offers fantastic prospects to realize use impact analyses. The purpose of the study is to examine the usage impact of memory, cache, storage, and bus on CPU performance using the Sugeno type and Mamdani type ANFIS models to determine the state of the computer system. The suggested method is founded on keeping an eye on how computer parts behave. The developed method can be applied for all kinds of computing system, such as personal computers, mainframes, and supercomputers by considering that the inference engine of the proposed ANFIS model requires only its own behavior data of computers’ components and the number of inputs can be enriched according to the type of computer, for instance, in cloud computers’ case the added number of clients and network quality can be used as the input parameters. The models present linguistic and quantity results which are convenient to understand performance issues regarding specific bottlenecks and determining the relationship of components.

## 1. Introduction

The performance of general-purpose computer systems is evaluated depending on the area of application of the nominal, complex, system, and workload performance. Nominal performance characterizes only the speed, or the performance, of the devices that make up the system. The complex performance considers not only the speed of devices but also the structure of the system—its effect on the speed of jointly functioning devices. System performance considers both the above factors—the performance of devices and the structure of the relationships between them, and the influence of the operating system. Performance on a workload displays all the factors affecting system performance, and, in addition, the properties of the workload—tasks that are solved by the computer system. Closely related to performance is a characteristic of the quality of user service, such as response time, i.e., the residence time of tasks in the system. Therefore, when evaluating performance, not only the number of operations executed by the system per unit of time is determined, but also the response time for the entire set of tasks and individual classes of tasks [1].

The performance of the computer system is related to the duration of the task processing processes, which depends on three factors: (1) workload; (2) system configurations; (3) task processing mode. These three factors together determine the order of development of computational processes over time, and the first task of analyzing performance is reduced to the dust of compact and informative forms of representation of computational processes. These forms create a conceptual basis for evaluating the functioning of computing systems during operation and during research using performance models. The second task of the analysis is the creation of models that allow predicting the performance of systems for various configurations, processing modes, and, possibly, different workloads.

Nowadays there are three main goals for assessing the performance of a computer system. These are choosing a computer for a user, designing a computer for a manufacturer, and improving configuration [2]. The purpose of these spheres is the optimization of the system’s response time and workloads. The evaluation of computer condition is to determine how well a certain system satisfies and correlates with current requirements and resources. Many researchers proposed a lot of approaches to evaluate the computer performance and prediction methods of hardware utilization, for instance, mathematical models for identification of computer state [3], and benchmark-based [4] and synthetical programs [5] have presented perfect results in the assessment of computer hardware components. However, these assessment techniques are based on some datasets that cause some inaccuracy during the implementation in the particular systems and on analyzing existing real systems. Additionally, there are fuzzy models for predicting the utilization of CPU based on trends and previous states of the workload [6,7] or taking into account historical data of the central processor time, RAM (read time, write time, swap time), the throughput of I/O and bus. In addition, these objects have many pointers (utilization, time, throughput, latency, delay, speed, frequency, and percentage), which confuses the assessing and prediction of computer components. However, all proposed methods are complex, despite the good result. These evaluation techniques have troublesome data acquisition for users and require programming knowledge, algorithms of machine learning, dataset manipulations for evaluation of computer state, and familiarity with hardware architecture. In addition, the methods of the above-discussed techniques are not clear, and the results of performance evaluation and utilization of hardware components are not comprehensible to all users.

Using the notion of fuzzy sets, we suggest transforming the data into linguistic variables in order to make all kinds of data more understandable and to employ unique metrics. The primary goal of the study is to develop an evaluation model using the adaptive neuro-fuzzy inference system (ANFIS) for all types of computing systems, including personal computers, mainframes, and supercomputers. The recommended model seeks to evaluate the computing system’s performance, forecast central processing usage by examining how hardware components interact with one another, and provide data for drawing conclusions about the hardware bottleneck and incompatibility.

Based on our prior research [8], we created new, improved ANFIS models in this study. In this study, we created Mamdani and Sugeno type ANFIS models to assess the state of the computer in terms of the interaction between its components. Meanwhile, in earlier work we employed two inputs for our fuzzy models, and the achieved results were not satisfied. To improve the model’s performance, we enhanced it by input objects and their datasets, which include cache, RAM, storage, and bus utilities. Particularly in Mamdani type ANFIS, adding two input components significantly enhanced the model evaluation outcome. The knowledge basis of the inference engine has also undergone some adjustments, as a result of which a sizable dataset and rule base have been added to it. Our ANFIS models produced more accurate results as a result.

### Related Works

Numerous researchers employed ANFIS for a variety of important research projects, including those in the fields of industry, finance, weather forecasting, health, and water quality prediction [9,10,11,12,13]. According to [14], air pollution (SO_2_) data collected at Bhadra station were combined with meteorological data to forecast the average air temperature using ANFIS. The wavelet’s approximately decomposed sub-series data are fed into the ANFIS. ANFIS and hybrid wavelet ANFIS (Gaussian membership) are contrasted (Gbel membership). The ANFIS (Gauss membership) displays superior results compared to the other two approaches, with a coefficient of determination (R2) = 0.95 and RMSE = 0.74. The effectiveness of the discrete wavelet transforms and ANFIS in time-series data modeling of weather characteristics is examined by Munandar [15]. The foundation of the statistical model is a linear regression plot of projected data outcomes. At the weather station at Indonesia’s Bungus port, data were checked every 10 min. When milling aluminum with a ball end, the authors of [16] used ANFIS to forecast surface roughness. After using a ball end mill, the study determined the average surface roughness Ra value for aluminum. ANFIS models have been used to forecast Ra in 84 studies. The ANFIS model, which is built with three Gaussian membership functions for each input variable and a linear membership function for the output, was chosen based on the minimal value of root mean square error (RMSE). The theoretical model and the response surface model have been compared to the ANFIS model (RSM). With the use of the ANFIS network and the features extracted from vibration signals, Helmi et al. [17] have proposed a structure for the diagnosis and detection of rolling bearing faults. Vibration signal defect information has been retrieved using time-domain and frequency-domain statistical properties. The findings show that this strategy outperforms other methods in terms of accuracy and classification performance. A thorough automatic system based on ANFIS is introduced by Awadallah et al. [18] for the detection of stator short circuits in brushless DC motors. A discrete-time numerical model was used to determine the performance of the system under typical operation. By adjusting the model to account for the fault current and accommodate the short circuit, a flawed performance was attained. The ANFIS for risk prediction for the computational grid environment is presented by Abdelwahab et al. [19]. The fuzzy model was set up by academics to evaluate the risk of the computer system’s expanding computing needs. The risk calculation of individual grid components, such as storage, CPU, memory, etc., is not taken into account by the authors in their work. In order to anticipate cloud workloads, Amekraz et al. [20] introduce a system known as CANFIS, which combines the Savitzky–Golay (SG) filter, chaotic time series analysis, and the adaptive neural fuzzy inference system (ANFIS). The SG filter is used to remove noise and outliers from the data, and chaotic analysis is used to look into how chaotic the workload is and to create the enhanced ANFIS model. Real workload traces from web apps, including NASA Kennedy and Wikipedia, are used to assess the suggested technique (i.e., CPU and memory of Google cluster). For the purpose of predicting CPU utilization, Bey et al. [21] combined ANFIS with the clustering procedure employed on CPU utilization time series. The more pertinent research is found in [6], where the authors applied fuzzy logic control theory to modify CPU workloads as needed such that use converges with a specified setting even for dynamic workloads. The study stated that fuzzy logic control is unrelated to a mathematical model of a controlled system or operating range, in contrast to proportional, integral, and differential (PID) approaches and other predictive models. The probability of introducing design flaws as a result of statistical inaccuracies included in the black box installation model is small because there is no fuzzy logic controller model. By conducting a logical study of the nonlinear relationship between utilization and load variations, Basaran and his research colleagues suggested a new fuzzy closed-loop system for managing CPU use [6].

Butt et al. [22] present the process scheduling ANFIS for the multi-tasking operating system. In their research, the simulator was designed in such a way that the packaging time, arrive time, and the final time of each completed process were used as input variables. The proposed simulator computes the last processor workload and utilization for each process using the defined formula. For each process, fuzzy dynamic priority is generated by the proposed ANFIS. According to the dynamic priority of each process in descending order, the process for execution of that process will be selected which will be first in the queue. The CPU design was proposed to be optimized in [23] using a unique fuzzy logic technique in order to reduce power consumption, boost productivity, measure and manage heat dissipation in the derived Pareto-optimal configurations, and more. By examining CPU design knowledge conveyed using fuzzy logic rules during the investigation of the design space from points of view of both the solution quality and speed of convergence, they proposed the improvement of microarchitecture using selective load value prediction (SLVP). In the studies of Chen et al. and Beghdad et al., we can see the C-means clustering methods. Chen et al. [7] present a fuzzy-based neural network for resource prediction, that is a self-adaptive prediction algorithm for resources of cloud computers. They combined subtractive fuzzy and C-means clustering methods to optimize convergence characteristics and training speed. To increase the reliability and performance in real time, the method is optimized using self-regulating training speed and weight of impulse. A CPU utilization predicting model was proposed by Beghdad et al. [24]. The model was built on the clustered instance selection in the previous stages. The proposed ANFIS uses a naive Bayes network controller for CPU prediction. A C-means clustering algorithm is used to estimate the direction of the next step. Hussain et al. [25] proposed a brand-new clustered induced ordered weighted averaging (CI-IOWA) Adaptive neuro-fuzzy inference system (ANFIS) model. This fuzzy time-series prediction model addresses the nonlinear relationship of the cloud QoS dataset and minimizes data dimensions. The suggested approach incorporates a fuzzy neural network structure for the best possible prediction outcomes and an intelligent sorting mechanism to control prediction uncertainty. The suggested method sorts input arguments based on associated order-inducing factors and then applies customized weights in accordance. This is carried out using the IOWA operator. Three fuzzy clustering techniques—fuzzy C-means (FCM), subtractive clustering, and grid partitioning—are used to further categorize the inputs. The inputs are then passed on to the ANFIS structure, which combines the advantages of neural and fuzzy networks.

In order to address concerns with over- and under-provisioning, research on resource consumption prediction in the cloud has been conducted by Malik et al. [26]. Over-provisioning of resources results in higher expenses and increased energy use. However, under-provisioning results in SLA violations and a decline in quality of service (QoS). The majority of the currently used mechanisms concentrate on the forecast of a single resource’s use, such as memory, CPU, storage, network, or servers assigned to cloud applications, but they ignore the correlation between resources. This study focuses on multi-resource consumption prediction utilizing a hybrid genetic algorithm (GA) and particle swarm optimization (PSO) functional link neural network (FLNN). The suggested method is assessed using Google cluster trace data. Hamid et al. [27] propose a fuzzy model for the load balancing method that provides fault tolerance by properly distributing the load according to user tasks among the currently available resources using anomalies and malfunction detection. By monitoring the current state of the system and the fairness in the distribution of tasks, the authors made calculations of priority value for each resource, i.e., fuzzy evaluation of CPU utilization, and tried to avoid overload problems that are the cause of the system renouncement. When anomalies are detected, the algorithm instructs the system to apply the failure rejuvenation mechanism to a virtual machine with abnormal behavior. Li et al. [28] proposed a fuzzy logic controller for load balancing. Their approach analyzes the correlation between several parameters such as CPU, memory, and I/O utilization that affect load balancing and uses the fuzzy logic algorithm, obtaining the load of several virtual servers. Then, it chooses the least loaded virtual server to handle the request and, if necessary, installs the server waiting or restart policy. This procedure involves checking the load state of virtual servers in real time. They used a software-defined networking (SDN) simulation platform for testing the correctness and effectiveness of this proposed algorithm.

Moreover, many researchers investigated CPU evaluation using the artificial neural network (ANN), as inaccurate resource predictions result in either low or high cloud resource provisioning. Valarmathi et al. [29] focused on finding a proactive solution to this issue. The majority of the prediction models now in use are based on a single workload pattern, making them unsuitable for handling unusual workloads. The researchers tackled this issue by employing a modern approach to dynamically evaluate CPU consumption in order to correctly estimate data center CPU utilization. For resource estimate, the suggested architecture uses deep architectural models based on ensemble random forest–long short-term memory. The data in this approach are preprocessed and trained using historical observation. An actual cloud data collection is used to analyze the strategy. According to the empirical interpretation, the new design beats the earlier strategies since it bears a 30–60% increase in resource usage accuracy. The evolutional neural network was proposed by Mason et al. [30]. To forecast CPU consumption, they made use of several distinct recurrent neural networks. Particle swarm optimization (PSO), differential evolution (DE), the covariance matrix evolutionary adaptation method (CMA-ES), and the dataset of PlanetLab files were employed for the suggested recurrent neural network. Duggan et al. [31] proposed a CPU utilization prediction approach based on machine learning algorithms. They also used an RNN and trained it in the Google cluster trace dataset using a backward propagation algorithm to predict the utilization of host CPU. Kumar et al. [32] proposed a workload prediction model for cloud data centers using the LSTM techniques. The model was tested in three datasets from the NASA HTTP server, Saskatchewan server, and Calgary server, and the result is incredible, i.e., it can perform a 60 min forecast of the workload of large data centers. Datrois et al. [33] developed a technique using machine learning approaches to forecast 24 h resource availability. The prediction technique uses quantile regression to provide an elastic relationship with a resource for restoration and the determination of unutilized resources. Computer components such as CPU, memory (RAM), storage, and network metrics were predicted to provide full availability. A study of SISO and MIMO models, including adaptive reliable controllers for the task of allocating CPU resources by virtual machines and satisfying certain QoS requirements, was presented in [34]. The controllers were aimed at adjusting CPU resources based on observations of previous CPU loads. In their work, the system takes into account only the CPU capacity, and the resource requirements are interconnected in several dimensions (i.e., calculations, storage, and network bandwidth). The identification/training of a system to extract interconnected information between resource requirements to consolidate workload when performing service level objectives is explored.

Many evaluation techniques have been proposed by researchers. The bulk of them have complicated metrics and parameters such as utilization, time, throughput, latency, delay, speed, frequency, and the percentage which are difficult to understand and use in the assessing process. According to these, we proposed a simplified evaluation method using components’ utilization in percentage scale and its linguistic values. In the implementation part of our idea, we chose a personal computer and we used its CPU, memory, cache, storage, and bus utilization. Many evaluation approaches of computer performance indicate that the main computer’s components which influence the performance of CPU are memory, cache, storage, and I/O devices. In contrast to approaches in the literature, the model shows language outcomes. In the future, performance counter correlations will aid in the creation of algorithms that can determine if a certain computer’s performance will be impacted by a given priority. The performance assertions generated from these approaches will allow resource management strategies to prevent performance degradation, allowing the infrastructure to operate safely and according to plan.

## 2. Proposed Idea

### 2.1. ANFIS

Automated control, decision-making, expert systems, data classification, and computer vision are some of the fields where adaptive neural-fuzzy inference systems have been successfully applied. Numerous academics use terms such as fuzzy rule-based systems, fuzzy expert systems, fuzzy modeling, fuzzy associative memory, fuzzy logic controllers, and just plain fuzzy systems to describe the numerous adaptive neural fuzzy inference systems that are currently available. Figure 1 depicts the general architecture of ANFIS.

In order to control linguistic aspects for the evaluation and prediction of CPU consumption, neuro-fuzzy was chosen. It is a good alternative for decision-making and for quality modeling of the human knowledge base. Artificial neural networks have the capacity to learn. Like humans, an ANN can notice patterns that can be used as input to provide predictions. Performance, error reduction, and adaptability can all be increased by integrating an ANN and fuzzy logic [13]. An adaptable neural network is used to build the fuzzy inference system, also known as the adaptive neuro-fuzzy inference system. ANFIS may use human knowledge, such as fuzzy if–then rules, and approximative membership functions, to produce an input–output mapping from the input–output data pairs supplied for the purpose of neural network training. An adaptive neuro-fuzzy inference system (ANFIS) is the process of creating a FIS using the framework of adaptive neural networks [13]. By using two different techniques, ANFIS learning updates the parameters of the membership function: backpropagation for all parameters and a hybrid technique that uses least-squares estimation for the parameters relating to the output membership functions and backpropagation for the parameters relating to the input membership functions. As a result, at least locally, the training error lowers as learning progresses. The ensuing parameters, which define the coefficients of each output equation, are found using the least-squares approach and a fuzzy rule base of the Sugeno type. The training process continues until the required root mean square error (RMSE) between the desired and generated output is attained, the right number of training steps (epochs), or both. The foundation and subsequent parameters of a first order Sugeno type fuzzy system for estimating and forecasting CPU utilization are established in this study using a hybrid learning technique.

### 2.2. Mamdani and Sugeno Type ANFIS

This work is a logical continuation of our previous work [8]. In this paper, we propose ANFIS which implements the principles and mechanisms of fuzzy set theory to assess the utilization of CPU by taking into account the influences of four components, cache, memory (RAM), storage, and bus throughput, on the performance of CPU while running multiple applications at the same time. In the previous work [8], we analyzed CPU utilization by two input components and the training dataset was not so big. Using the performance monitor (perfmon.exe) of the operating system, we collected data of components’ utilization, such as processor time, processor utility, paging, disk time, cache time (by retrieving the data directly from the cache), and bus utility, respectively, and information about CPU, RAM, storage, cache, and bus utilization. In the monitoring period, we created a utilization dataset of memory, cache, storage, bus, and CPU workloads of the test bench computer. The training dataset is illustrated in Table 1.

An adaptive neuro-fuzzy inference system that uses a rule base system was created by human experience and knowledge [35]. In this paper, to apply our idea we developed Sugeno type ANFIS and Mamdani type ANFIS models for assessing the utilization value of CPU, by evaluating the effect of RAM, cache, storage, and bus utilizations on CPU usage. There are some differences between the Sugeno and Mamdani ANFIS, i.e., they have a different knowledge base and outputs are varied. Sugeno type ANFIS generates the knowledge base by a training dataset and displays the numerical information about the CPU workload, and Mamdani type ANFIS has the rule base knowledge and outputs a linguistical assessment of the CPU utilization status, for example, low, middle, or high. Both ANFIS models contain the same modules, i.e., fuzzification, knowledge bases, inference engine module, and defuzzification modules [8]. Although Sugeno ANFIS and Mamdani ANFIS have the same modules, their constitution is different, i.e., the internal mechanism of the fuzzy inference system is different according to the rule base and training dataset. Figure 2 demonstrates the differences of the two models.

We used the algorithm for our ANFIS which is presented in Algorithm 1.
**Algorithm 1:** Steps of ANFIS algorithm1: Defining of linguistic variables for each hardware component2: Constructing membership functions for each linguistic variable3: Developing knowledge base (rule base for Mamdani ANFIS and training dataset for Sugeno ANFIS)4: Fuzzifying the crisp inputs5: Training process and evaluating knowledge base (database, dataset, and rule base)6: Combining the output results of each rule7: Defuzzifying nonfuzzy outputs

An adaptive neuro-fuzzy inference system starts the process from the fuzzification module, but before that, linguistic variables and membership functions must be defined. In the fuzzification module, all the RAM, cache, storage, and bus utilization numeric values of inputs are fuzzified into fuzzy inputs. In this step, we utilized the Gaussian type membership function and the function is shown in Equation (1).
(1)$$gaussian\left(x;c,\sigma \right)=e^{-\frac{1}{2}{\left(\frac{x-c}{\sigma }\right)}^{2}}$$

In Equation (1), *c* represents the center of the membership function, and *σ* determines the width of the membership function [35]. For instance, we defined a “middle” linguistic term by the Gaussian function in the range of (*x*: 50, 30). It means that 50 is the center of the membership function and here the weight of the function obtains the highest quality of 1, and accordingly, when the membership function reaches 30, the value of the function will be 0.5. For building the membership function of variables for all inputs and outputs, we used Equation (1).

The module of the inference engine produces outputs or predicts the system results by applying the dataset and rule base. The inference engine module of the proposed ANFIS was built on an “*if–then”* rule set [35]. As already mentioned, we propose two kinds of adaptive neuro-fuzzy inference systems. These are Mamdani ANFIS and Sugeno ANFIS.

The Mamdani type ANFIS which we offer is implemented by using the following steps:Formulating a list of fuzzy rules.Using membership functions to fuzzify the input values that are crisp.Combining inputs that have been fuzzified in accordance with fuzzy rules to determine the rule strength.By combining rule strength and output, determining the rule’s effect.Combining the outcomes of obtaining an output distribution.Defuzzification of the results.

According to the combination of four inputs’ and one output’s linguistic variables, we developed 81 rules for Mamdani ANFIS and one example of the rules is described in Equation (2). Based on these 81 rules, the inference engine of Mamdani ANFIS generates the rule strength. The equation shows that RAM, cache, storage, and bus are inputs. Their crisp values are fuzzified by the Gaussian membership function and using the logic operator “AND” combines all four inputs to obtain the rule strength.
(2)$$IF\left(RAM\;is\;Low\;and\;Cache\;is\;Low\;and\;Storage\;is\;Low\;and\;Bus\;is\;High\right)\;THEN\;\left(CPU\;is\;Low\right)$$

The inference engine module of Sugeno ANFIS is different, i.e., this module generates its rule strengths according to the training process. In the training step, the inference engine using the backpropagation method builds optimal weights to predict the utilization of CPU. For the training process, we used the dataset which is illustrated in Table 1. The detailed calculation process for rule strength generation of the Mamdani ANFIS and Sugeno ANFIS is shown in Figure 3.

The defuzzification module is the final step of ANFIS, it transforms the results of fuzzy outputs, i.e., according to the Mamdani type ANFIS it provides linguistic values and in Sugeno type ANFIS it provides crisp values. In this step, the input numeric values are fuzzified, the rule strengths are implemented, inference engine is trained, and rule strengths built. Here, rule strengths compute fuzzy output data and transform them into numeric values. For Mamdani ANFIS, we used the centroid defuzzification method which is the default and for Sugeno type ANFIS we used the weighted average method, as illustrated in Figure 3.

## 3. Data Acquisition and Performance Evaluation

### 3.1. Data Acquisition Application

The task of the system monitor is to log the internal states of the computer system. The information obtained by system monitors allows us to solve problems in a wide range of applications, for example, detecting some errors in a computer system, checking resource usage and workload of hardware components, providing a basic rule for building models of a computer system, and finding bottlenecks in the system. Using the system monitor, we acquired data for our ANFIS models. For data acquisition, we chose a personal testbed computer. In Table 2, the testbed computer’s specific parameters are presented.

As explained in Section 2, in our testbed computer we simultaneously ran multiple applications and monitored the objects mentioned below and their aspects, then collected the data described in Table 1. Monitoring CPU directly measures the CPU workload and utilization. Due to the dependence of multi-processor systems on such metrics, they are particularly crucial. That is why it is necessary to regularly examine the system for active and idle processors, as well as which ones are capable of completing the pending task. The “Memory” section displays the current state of memory, we monitored “Pages/sec” to obtain data about memory utilization. The way a cache behaves affects a computer system’s overall performance. One can deliberately configure the system if the real behavior of the system is known. The communication between system components is centered on the buses. An indicator of system component activity is bus activity. We can infer information about bus protocol problems, the kind and volume of messages in the bus, and system bottlenecks by measuring the bus throughput. More effective indicators for obtaining storage utilization and workload are monitored, and these are “Average transfer rate”, “Disk time”, and “Disk queue length” [8].

### 3.2. Developing the ANFIS for Evaluation of CPU Utilization

Five layers make up the ANFIS structure: the fuzzy layer, the product layer, the normalized layer, the defuzzy layer, and the overall output layer. The development process of these five layers of ANFIS consists of three steps, these are:Fuzzification of input RAM, cache, storage, and bus values—fuzzy layer.Determination inference method and rules (data)—product and normalized layers.Defuzzification of CPU utilization values as output—defuzzy and output layers.

In the fuzzification step, we used the Gaussian membership function to fuzzify input variables, i.e., in this step we measured percentage quantity of RAM, cache, storage, and bus utilizations for linguistic variables. We used the same linguistic variable distribution and Gaussian membership function for all components, as shown in Table 3.

In the next step, according to the knowledge base we defined two kinds of inference methods, as shown in Figure 4. One is for Mamdani ANFIS based on the rule base and the second one is Sugeno ANFIS based on the dataset of influences among the resource utilizations.

The determination inference method and rules i.e., creation of normalized layers, are different. The normalized layer of Mamdani ANFIS is based on an “if–then” rule base. The rule base of Mamdani ANFIS is linguistic and developed according to utilization data of hardware components. The rule base is developed by analyzing the impact of four input resource utilizations to one output CPU utilization with their three linguistic variables and consists of eighty-one rules. The rule development process is described in Figure 5. This rule base determines the inside layer of the ANFIS model. If multiple rules are active for a single membership output function, only one membership value must be selected. This procedure is called fuzzy conclusion or fuzzy solution. Mamdani type ANFIS finds the association among the variables for assessing the ANFIS [35]. The formula in Equation (3) is used for the centroid defuzzification method in Mamdani type ANFIS.
(3)$$Z=\frac{\int \mu_{c}\left(z\right)\times zdz}{\int \mu_{c}\left(z\right)dz}$$

In the Sugeno type ANFIS, we used the weighted average defuzzification method and Equation (4) describes it [35]:(4)$$Z=\frac{\sum \mu_{c}\bar{Z}\times \bar{Z}}{\sum \mu_{c}\bar{Z}}$$

The normalized layer of Sugeno type ANFIS is developed by training the dataset. The training process of Sugeno ANFIS determined the weights of layers. The training data come from the application of a system monitor, as shown in Table 1. The fuzzy logic toolbox is provided with an ANFIS editor window, and it is designed for training Sugeno type ANFIS logical output mechanisms. We used a hybrid algorithm for the training process, which includes two methods, that is, the gradient descent method and the least-squares method. The training process of Sugeno ANFIS is completed in 1000 epochs, and accuracy is 0.231 as described in Figure 6. The obtained training process accuracy of our Sugeno ANFIS model is significantly suitable, especially when we have multiple inputs.

Thus, we completed the rule base, inference method determination, and defuzzification steps of our ANFIS models. In Figure 7, the Rule Viewer shows the input parameters of Mamdani type ANFIS and Sugeno type ANFIS where RAM = 95.6, cache = 94.4, bus = 95.0, and storage = 60.0. The Sugeno ANFIS evaluated CPU utility as 19.3. Mamdani ANFIS assessed it as 19.5 by following the established rule base. The linguistic result of “Low” is equivalent according to the membership function distribution. In the ANFIS Rule Viewer, we can perform the evaluation as shown in Figure 8.

## 4. Experimental Results

The total amount of work a central processing unit handles is known as CPU usage. The performance of the system is also estimated using it. The volume and kind of computing jobs might affect CPU utilization because some take a lot of CPU time while others do not. In this paper, we focus on an experiment on system performance and RAM, cache, storage, bus, and CPU utilization. This section contains the evaluation results of the simulated ANFIS models. The performances of ANFIS models are demonstrated by comparing their results with actual CPU utilization status which was acquired from a system monitoring application. The system monitoring application shows the numerical view of the actual states of computer resources in real-time mode. The comparison of results of ANFIS models with the system monitor’s data allowed us to evaluate the reliability of our ANFIS model.

During the hundred seconds, we monitored the state of computer components’ indicators, such as processor utility (CPU), disk time (storage), cache, bus throughput, and pages (RAM). Simultaneously, the obtained data were assessed using the Mamdani and Sugeno type ANFIS models. In the first five columns of Table 4, we present the system monitor application data. The table also presents the quantity and linguistical values of Mamdani and Sugeno ANFIS models in the remaining four columns. Both ANFIS models’ results are practically similar to the actual utilization state of CPU. It means that our developed ANFIS models evaluated the utilization status of CPU reliably, that is, the inference engine containing the component rule base and training dataset was successfully studied by the output mechanisms of ANFIS. The linguistical results of the Mamdani type ANFIS model are more understandable and demonstrate a clear visual conclusion about CPU utilization.

The graphical visualization of performance monitor data shows the actual state of the cache, RAM, storage, bus, and CPU utilization, Figure 8. According to these graphics, we can analyze the impact of the above-mentioned components on CPU utilization. To carry out analyses about the influence, we represented whole components’ influence on CPU and the particular influence of each component on CPU utilization. As we see in the graphs, it is difficult to realize fully linear characteristics between the CPU and particular components, but each hardware component in many cases has a linear impact on CPU. In Figure 9 and Figure 10, cache and RAM are more important, and the results show that their utilizations have more impact on CPU utilization on each level, i.e., increasing and decreasing the utilization level significantly impacts CPU. In Figure 11 and Figure 12, we can see that storage and even bus have linear characteristics when utilization is quite low. Moreover, the whole influence of all components is clearly represented in Figure 8 and the total utilization level of all components has more sharp and linear characteristics in CPU utilization. The results of the analyses demonstrate that the tested computer hardware has linearity and that each hardware component is compatible with the others. System compatibility indicates that no computer reconfiguration or redesign is required; instead, the computer system handles and manages the actual workload.

To evaluate the proposed ANFIS models, we compared the ANFIS data with actual data about CPU utilization. The achieved results are impressive despite some incompatibilities. The results of the ANFIS models are shown in Figure 13 and Figure 14. As aforementioned, we used linguistic variables and we defined their distribution diapasons. Table 5 shows error prediction results, and average accuracy in the “Middle” diapasons is 0.16 for our proposed Mamdani and Sugeno ANFIS models. In “Low” and “High” distribution diapasons, both models show a difference from the actual status of CPU, especially in the “High” level. The Mamdani type ANFIS model’s maximum and minimum error range is between 4.3 and −4.8. The maximum and minimum error indicator of Sugeno type ANFIS is between 5.6 and −3.1. Sugeno ANFIS has more accurate results in the “Low”, “Middle” levels. In the “Low”, “Middle”, and “High” linguistic levels, Sugeno ANFIS average error results are, respectively, −0.05, −0.18, and −2.26. Despite these weaknesses, we assess that the proposed ANFIS models achieved significant evaluation results.

## 5. Conclusions

The proposed ANFIS models for evaluating processor performance are based on specific parameters, which generally represent common parameters, such as the average percentage of the workload associated with cache, RAM, bus, and storage. We assumed that the utilization of cache, memory, storage, and bus includes all the main features affecting the infrastructure of the computing system, namely the CPU utilization. We used adaptive neuro-fuzzy logic to determine the evaluation rule strength by enhancing our previous work [6]. This ANFIS is useful for any user who has an elementary knowledge and is just studying performance and utility of computer components. Computational experiments were carried out based on adaptive neuro-fuzzy techniques to evaluate the utilization influence of cache, RAM, bus, and storage on the state of CPU. For input components, we determined linguistic variables such as “Low”, “Middle”, and “High”. Experimental results allow us to analyze hardware components of the computer in linguistic mode, that is, in the form of human language, which improves the judgement for experts and impressively accelerates the decision-making process. In our work, the models’ results were analyzed by comparing the results of a testbed computer component’s actual state, and the rule strengths were built on this testbed computer, that is, the main concept of our ANFIS models is that they require their own behavior dataset. The knowledge base module partition dataset and rule base cannot be utilized in ANFIS models to evaluate the other computers’ states. To prove the reliability of the proposed ANFIS models, we made result comparisons only with the data acquired from the system monitor application. The comparison shows each ANFIS has different accuracy in each linguistic distribution. Mamdani type ANFIS predicts in the “Middle” level, near to the actual state, i.e., its average accuracy is 0.16. In the “Low” and “High” linguistic distribution, ANFIS prediction error reaches, respectively, 1.25 and −3.86. The Mamdani type ANFIS model’s maximum and minimum error range is between 4.3 and −4.8. The maximum and minimum error indicator of Sugeno type ANFIS is between 5.6 and −3.1. Sugeno ANFIS has more accurate results in the “Low”, “Middle” levels. In the “Low”, “Middle”, and “High” linguistic levels, the Sugeno ANFIS average error results are, respectively, −0.05, −0.18, and −2.26. Since the system monitor application provides only quantitative data about utilization of computer components and applications, it does not carry out any influence analysis among the hardware components. Unlike the system monitor, our Mamdani and Sugeno type ANFIS models receive utilization data and then assess CPU utilization. If actual state data of the system monitor and ANFIS outputs are close or similar to each other, then we can assess that the proposed models are reliable and we can use them for prediction and evaluation of CPU state and determine component compatibility. The proposed technique can be implemented in personal computers, mainframes, supercomputers, cloud computers, centralized computers, and distributed computers by using their own behavior dataset and rule base. In our future works, we are planning implement and test our model in the above-mentioned computers.

## Figures and Tables

**Figure 1 sensors-22-09502-f001:**
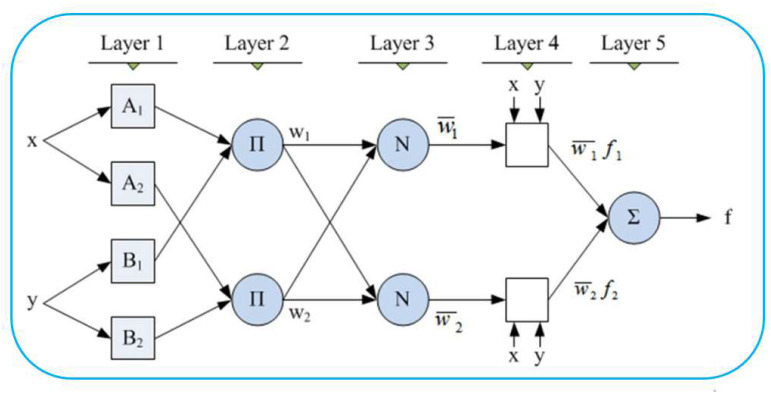
The architecture of ANFIS.

**Figure 2 sensors-22-09502-f002:**
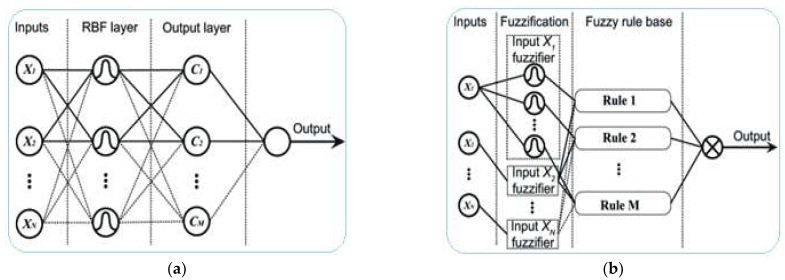
Inference engine layers of ANFIS models: (**a**) Sugeno ANFIS, (**b**) Mamdani ANFIS.

**Figure 3 sensors-22-09502-f003:**
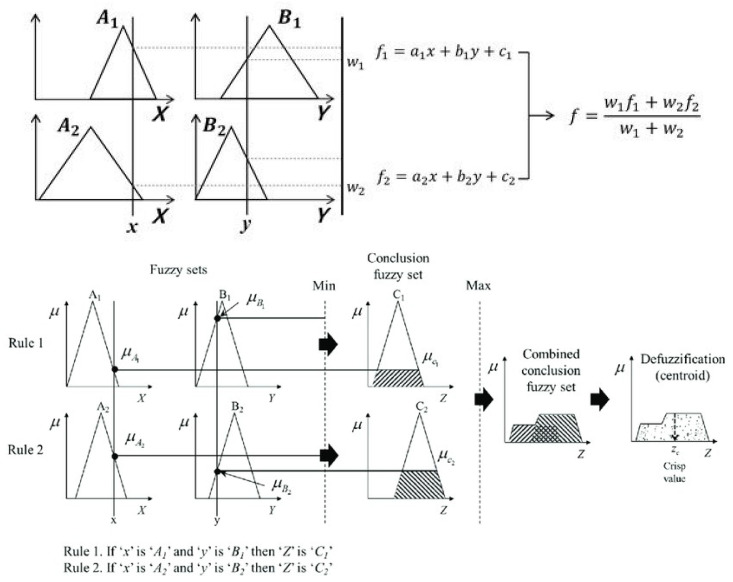
Rule base generation process of ANFIS models.

**Figure 4 sensors-22-09502-f004:**
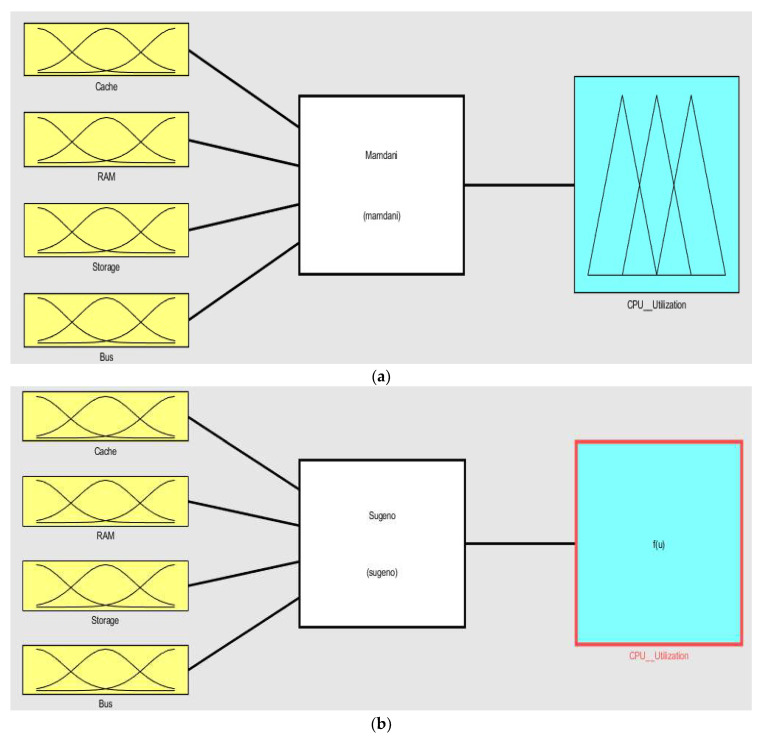
Input and output objects: (**a**) Mamdani type ANFIS, (**b**) Sugeno type ANFIS.

**Figure 5 sensors-22-09502-f005:**
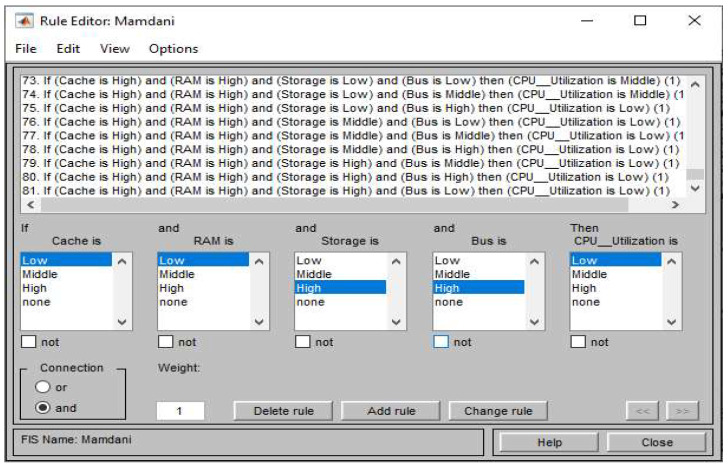
Rule base determination for inference engine of Mamdani ANFIS.

**Figure 6 sensors-22-09502-f006:**
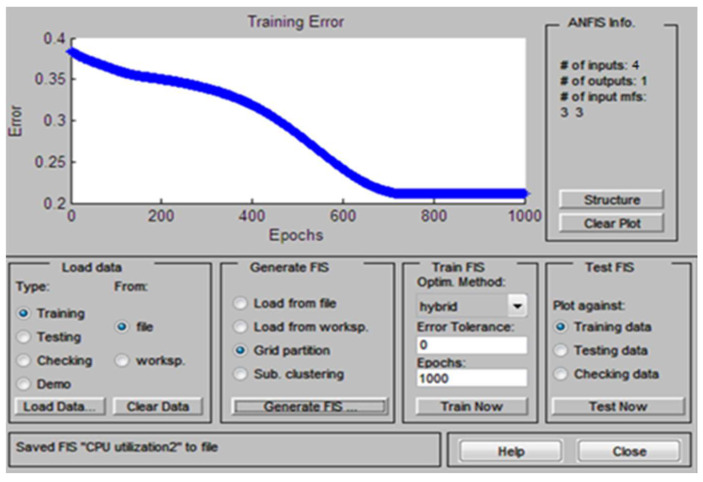
Training result of Sugeno ANFIS.

**Figure 7 sensors-22-09502-f007:**
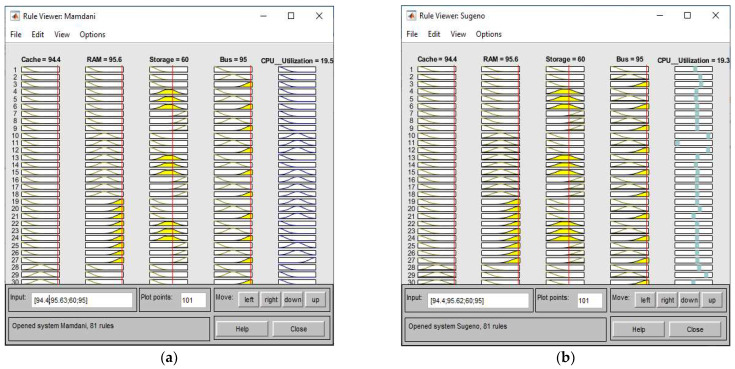
Rule Viewer output parameters for: (**a**) Mamdani ANFIS, (**b**) Sugeno ANFIS.

**Figure 8 sensors-22-09502-f008:**
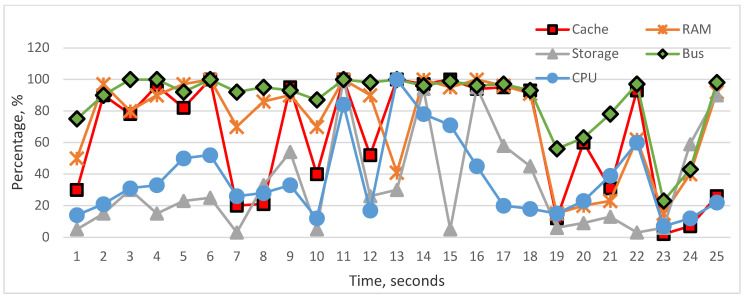
Utilization status of the components.

**Figure 9 sensors-22-09502-f009:**
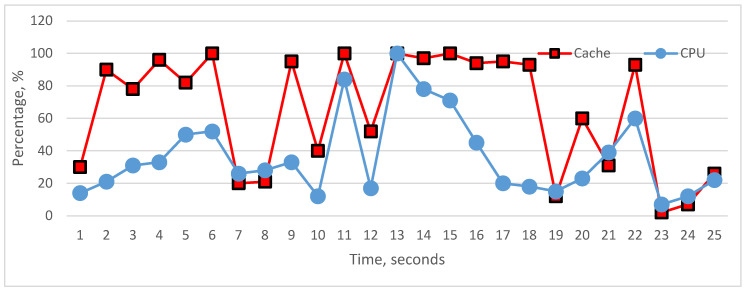
Utilization status of cache and CPU.

**Figure 10 sensors-22-09502-f010:**
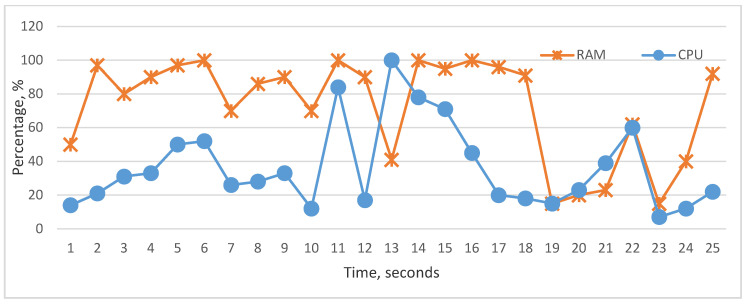
Utilization status of RAM and CPU.

**Figure 11 sensors-22-09502-f011:**
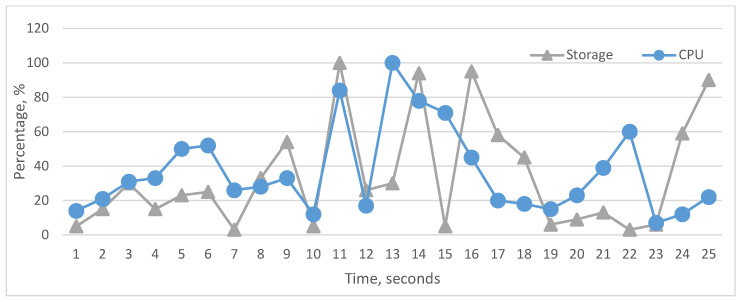
Utilization status of storage and CPU.

**Figure 12 sensors-22-09502-f012:**
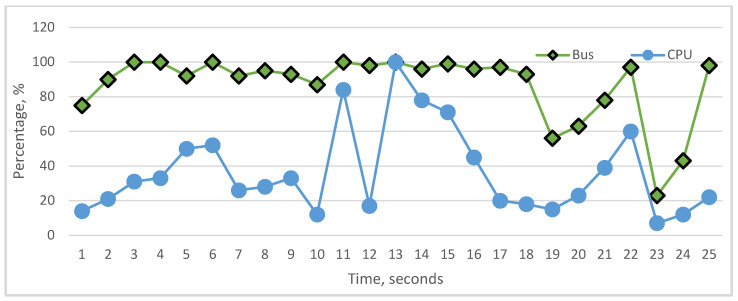
Utilization status of bus and CPU.

**Figure 13 sensors-22-09502-f013:**
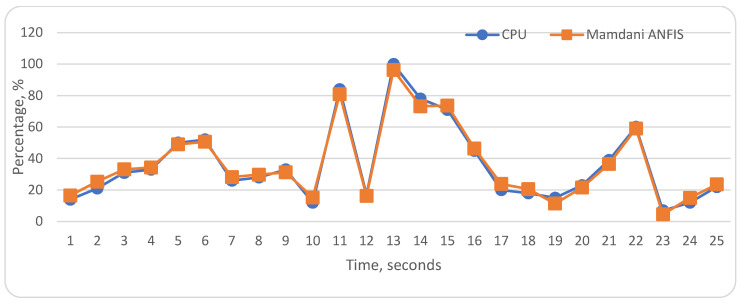
Actual CPU and Mamdani ANFIS evaluation.

**Figure 14 sensors-22-09502-f014:**
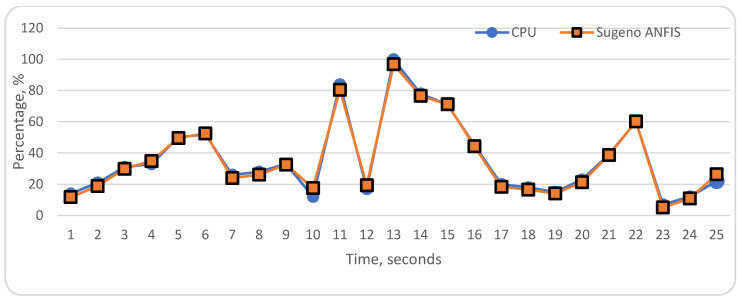
Actual CPU and Sugeno ANFIS evaluation.

**Table 1 sensors-22-09502-t001:** Training dataset.

№	Utilization (%)				
	RAM	Cache	Storage	Bus	CPU
1	51	32	5	72	14
2	38	97	25	100	35
…	…	…	…	…	…
3000	48	80	20	95	65

**Table 2 sensors-22-09502-t002:** Parameters of testbed computer.

Hardware and Software	Parameter
Operating system	Windows 10 Professional (Microsoft Corparation), 64 bit
Processor	Intel(R) Core(TM) i7-8700 CPU @3.20 GHz 3.19 GHz
Memory (RAM)	16.0 GB
Cache	L1 384 KB, L2 1.5 MB, L3 12.0 MB
Storage	1050 GB

**Table 3 sensors-22-09502-t003:** Linguistic variables and distribution diapasons.

Linguistic Variables	Distribution
Low	(0, 20, 40)
Middle	(30, 50, 70)
High	(60, 80, 100)

**Table 4 sensors-22-09502-t004:** Utilization data and ANFIS evaluation results.

Performance Monitor Evaluation (Percentage Utilization of Components)					CPU Evaluation of Mamdani FIS		CPU Evaluation of Sugeno FIS	
Cache	RAM	Storage	Bus	CPU	Crisp Value (%)	Linguistic Value	Crisp Value (%)	Linguistic Value
30	50	5	75	14	16.5	Low	11.8	Low
90	97	15	90	21	25.3	Middle	18.9	Low
78	80	30	100	31	33.1	Middle	29.9	Middle
96	90	15	100	33	34.3	Middle	34.9	Middle
82	97	23	92	50	49.0	Middle	49.6	Middle
100	100	25	100	52	50.6	Middle	52.6	Middle
20	70	3	92	26	28.2	Middle	24.0	Low
21	86	33	95	28	29.7	Middle	26.1	Middle
95	90	54	93	33	31.2	Middle	32.6	Middle
40	70	5	87	12	15.2	Low	17.6	Low
100	100	100	100	84	80.9	High	80.5	High
52	90	26	98	17	16.2	Low	19.4	Low
100	41	30	100	100	96.3	High	96.9	High
97	100	94	96	78	73.2	High	76.6	High
100	95	5	99	71	73.6	Middle	71.3	Middle
94	100	95	96	45	46.4	Middle	44.3	Middle
95	96	58	97	20	23.8	Low	18.3	Low
93	91	45	93	18	20.6	Low	16.6	Low
12	15	6	56	15	11.4	Low	14.1	Low
60	20	9	63	23	21.6	Low	21.3	Low
31	23	13	78	39	36.6	Middle	38.7	Middle
93	62	3	97	60	59.1	Middle	60.2	Middle
2	15	6	23	7	4.6	Low	5.1	Low
7	40	59	43	12	15.0	Low	10.9	Low
26	92	90	98	22	23.6	low	26.4	Low

**Table 5 sensors-22-09502-t005:** Prediction error and ANFIS evaluation results.

Performance Monitor		Mamdani ANFIS		Sugeno ANFIS	
Linguistic Value	Actual CPU State	Error	Average Error	Error	Average Error
Low	7	−2.4	1.25	−1.9	−0.05
	12	3.2		5.6	
	12	3		−1.1	
	14	2.5		−2.2	
	15	−3.6		−0.9	
	17	−0.8		2.4	
	18	2.6		−1.4	
	20	3.8		−1.7	
	21	4.3		−2.1	
	22	1.6		4.4	
	23	−1.4		−1.7	
Middle	26	2.2	0.16	−2	−0.18
	28	1.7		−1.9	
	31	2.1		−1.1	
	33	1.3		1.9	
	33	−1.8		−0.4	
	39	−2.4		−0.3	
	45	1.4		−0.7	
	50	−1		−0.4	
	52	−1.4		0.6	
	60	−0.9		0.2	
	71	2.6		0.3	
High	78	−4.8	−3.86	−1.4	−2.26
	84	−3.1		−3.5	
	100	−3.7		−3.1	

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
