# Peer review of "Computer State Evaluation Using Adaptive Neuro-Fuzzy Inference Systems"

_sensors, 2022, doi:10.3390/s22239502_

Round 1

Reviewer 1 Report

This paper titled “Computer State Evaluation using Adaptive Neuro Fuzzy Inference Systems” introduces a simplified evaluation method using components’ utilization in percentage scale and its linguistic values. Fuzzy set theory and adaptive neuro-fuzzy inference system (ANFIS) model presents great opportunities to realize the influence analyzations between the components' utilization. The aim of the paper is to determine the state of the computer system using the Sugeno type and Mamdani type ANFIS models to assess the utilization influence of memory, cache, storage, and bus on CPU performance. However, still, authors should cope with the following major reviewer’s concerns.

1.    It is hard to understand the main contributions in the introduction section of the manuscript.

2.    The authors have not identified the novelty nor described how the Adaptive Neuro-Fuzzy Inference System (ANFIS) is different from existing work.

3.    The authors have mentioned only one recent technique for Robust Dynamic CPU Resource Provisioning in Virtualized Servers proposed in 2020. All other related works mentioned are published before 2017and 2018.

4.    The mathematical modeling of the Adaptive Neuro-Fuzzy Inference System (ANFIS) is extremely simple and adopted from existing work without referring to them, which should be referred to.

5.    The evaluated metrics for the proposed ANFIS models in the results sections are the same as in the previous works (more or less), it seems the authors have reproduced/remodeled the existing work without elaborating on their contributions, which should be identified.

6.    The conclusion section of the manuscript needs a complete revision and more explanation.

7.    The authors have not mentioned any future works or directions of this research in the manuscript.

Author Response

Point 1: It is hard to understand the main contributions in the introduction section of the manuscript.

Response 1: I agree. We apologize for bad expression and semantic mistakes. We made some improvements which are highlighted in the whole text. In subsection 1.1. we tried to explain the motivation and from line 74th till 108th. We added more in detail information about the desired model. We tried to edit the whole text to tie the logical meaning of the paper, including the methods, experiments, results, and conclusion.    

Point 2: The authors have not identified the novelty nor described how the Adaptive Neuro-Fuzzy Inference System (ANFIS) is different from existing work.

Response 2: We highly appreciate your scientific attention and your deep analytical point. After your reviewing in the first round the scientific and semantic sides of our paper is observable improved. In this paper, we proposed simplified method for performance evaluation by analyzing the influences of components utilization. The simplicity of method is, that the models provides linguistic results, do not require big data, and deep knowledge, i.e., the small number of rule base are enough for evaluation and predicting the system status. Rule base generation process is by using linguistic distribution of utilization status. The prediction accuracy table is presenting the reliability of model. As we know, the mentioned techniques which are surveyed in the paper are not simple as ours, i.e., they use multiple metrics and parameters.

Point 3: The authors have mentioned only one recent technique for Robust Dynamic CPU Resource Provisioning in Virtualized Servers proposed in 2020. All other related works mentioned are published before 2017and 2018

Response 3: I agree. In this part, we analyzed and described in detail six newer papers. We added them to related works subsection by authors Dumitrescu et al. [10], Karnavel et al. [11], Amekraz et al. [18], Hussain et al. [25], Malik et al. [26], and Valarmathi et al. [29]. All the added papers are highlited in the manuscript.

Point 4: The mathematical modeling of the Adaptive Neuro-Fuzzy Inference System (ANFIS) is extremely simple and adopted from existing work without referring to them, which should be referred to.

Response 4: I agree and apologize for mistake. We highly appreciate your scientific attention. We updated the manuscript by refferring to [6], [13], and [35]. They are highlited in the manuscript.

Point 5: The evaluated metrics for the proposed ANFIS models in the results sections are the same as in the previous works (more or less), it seems the authors have reproduced/remodeled the existing work without elaborating on their contributions, which should be identified.

Response 5: The main differences are in the input of our models that is, cache, RAM, storage, and bus utilities, while in previous work we used two inputs for our fuzzy models. Adding two input components are remarkably improved the model evaluation result, especially in Mamdani type ANFIS. In addition, the structures of current models have some changes, i.e., the knowledge base of the inference engine is significantly enriched with a large dataset and rule base. Thus, we obtained more precise results from our ANFIS models.

Point 6: The conclusion section of the manuscript needs a complete revision and more explanation.

Response 6: I agree with your recommendation. As we mentioned in the paper, we experimented in personal computer and compared the results of our models with actual state of CPU which obtained via Performance monitor application. The application provides with real time and actual data about components. In the future works we are planning implement and experiment our models in cloud computers or in supercomputers and compare the results with other evaluation techniques.

Point 7: The authors have not mentioned any future works or directions of this research in the manuscript.

Response 7: I agree with your mention. As we mentioned in the paper, In the future works we are planning implement and experiment our models in cloud computers or in supercomputers and compare the results with other evaluation techniques.

Reviewer 2 Report

The authors present an interesting problem concerning the computer state evaluation using adaptive neuro fuzzy inference systems. They propose an evaluation based on components utilization in percentage scale. They are also represented as linguistic values.  Adaptive neuro fuzzy inference system (ANFIS) can be used to examine the influence analyzations between the components utilization. The authors decided to use Sugeno type and Mamdani type ANFIS in the research. They were used to assess the influence of utilization of memory, cache, storage and bus on CPU performance. Moreover, authors declare that propose solution can be used to assess different types of computing systems. The model shows how to understand the performance issues regarding the specific bottlenecks and how to determine the relationship between components

Presented paper is interesting and the problem is well-described. However some shortcomings must be improved:

* Line 92 -> additional gap between "base." and "Thus"

* The resolution of the Figure 1 could be improved; text is blurred in the currrent version

* Some sentences do not have logical sense -> Line 244/245 "In Table 1 illustrated the created dataset" - In should be removed or the sentence should be rewritten. These type of mistakes should be revised

* Figure 2 -> diagrams have different notations - first: (a); second: b) 

* reference to fuzzy inference system in text should be unified: one time there is an upper case reference, other time lower case or mixed -> this should be unified in manuscript

* Line 258 -> "these is ilustrated in Figure 2" should be rewritten

* Line 322 -> section 2 -> Section 2

* Fontsize in Figure 4 could be bigger, as currently it is hard-to-read

* Figures 8-14 should be improved: labels for x and y axis should have some padding regarding the axis, linewidths and markers size could be slightly lowered

The research presents quite an interesting problem directed into ANFIS and computer state evaluation. However, to accept this paper, the text should be checked to perform changes and the shortcomings listed above should be applied.

Author Response

Point 1: Line 92 -> additional gap between "base." and "Thus"..

Response 1: I agree. We apologize for bad expression and semantic mistakes.

Point 2: The resolution of the Figure 1 could be improved; text is blurred in the currrent version.

Response 2: I agree. The figure is improved.

Point 3: Some sentences do not have logical sense -> Line 244/245 "In Table 1 illustrated the created dataset" - In should be removed or the sentence should be rewritten. These type of mistakes should be revised.

Response 3: I agree. We apologize for bad expression and semantic mistakes. We made some improvements which are highlighted in the whole manuscript.

Point 4: Figure 2 -> diagrams have different notations - first: (a); second: b).

Response 4: I agree. We apologize for semantic mistakes. We corrected it.

Point 5: reference to fuzzy inference system in text should be unified: one time there is an upper case reference, other time lower case or mixed -> this should be unified in manuscript.

Response 5: I agree. Mentioned word is unified. They are highlighted in the manuscript.

Point 6: Line 258 -> "these is ilustrated in Figure 2" should be rewritten.

Response 6: I agree. We rewrited it and it is highlited in 430th line.

Point 7: Line 322 -> section 2 -> Section 2.

Response 7: I agree. It is corrected.

Point 8: Fontsize in Figure 4 could be bigger, as currently it is hard-to-read.

Response 8: I agree. We increased the size of figures.   

Point 9: Figures 8-14 should be improved: labels for x and y axis should have some padding regarding the axis, linewidths and markers size could be slightly lowered.

Response 9: I agree. We increased the size of figures and decreased the thickness of lines. 

Round 2

Reviewer 1 Report

All the recommendations are incorporated. Therefore, I would like to accept this manuscript for publication with minor English spelling and Grammar check. 

Regards,

Dr. Iqbal Qasim